From Lucy to Kadanuumuu: balanced analyses of Australopithecus afarensis assemblages confirm only moderate skeletal dimorphism

Reno Philip L. 1 philreno@psu.edu
Lovejoy C. Owen 2
1 Department of Anthropology, The Pennsylvania State University , University Park, PA , USA
2 Department of Anthropology and School of Biomedical Sciences, Kent State University , Kent, OH , USA
Jungers William
Electronic publication date: 2015 Apr 28
Publication date: 2015
Volume: 3
Electronic Location ID: e925
Received 2015 Feb 24; Accepted 2015 Apr 9
Copyright: © 2015 Reno and Lovejoy
Copyright year: 2015
Copyright holder: Reno and Lovejoy
License: This is an open access article distributed under the terms of the Creative Commons Attribution License, which permits unrestricted use, distribution, reproduction and adaptation in any medium and for any purpose provided that it is properly attributed. For attribution, the original author(s), title, publication source (PeerJ) and either DOI or URL of the article must be cited.
License URL: https://creativecommons.org/licenses/by/4.0/

Keywords: Sexual selection, Reproductive behavior, Homin, Human evolution, Hominid, Chimpanzee, Gorilla, Sexual dimorphism

Funding: The authors declare there is no funding for this work.

==============================
Sexual dimorphism in body size is often used as a correlate of social and reproductive behavior in Australopithecus afarensis. In addition to a number of isolated specimens, the sample for this species includes two small associated skeletons (A.L. 288-1 or “Lucy” and A.L. 128/129) and a geologically contemporaneous death assemblage of several larger individuals (A.L. 333). These have driven both perceptions and quantitative analyses concluding that Au. afarensis was markedly dimorphic. The Template Method enables simultaneous evaluation of multiple skeletal sites, thereby greatly expanding sample size, and reveals that A. afarensis dimorphism was similar to that of modern humans. A new very large partial skeleton (KSD-VP-1/1 or “Kadanuumuu”) can now also be used, like Lucy, as a template specimen. In addition, the recently developed Geometric Mean Method has been used to argue that Au. afarensis was equally or even more dimorphic than gorillas. However, in its previous application Lucy and A.L. 128/129 accounted for 10 of 11 estimates of female size. Here we directly compare the two methods and demonstrate that including multiple measurements from the same partial skeleton that falls at the margin of the species size range dramatically inflates dimorphism estimates. Prevention of the dominance of a single specimen’s contribution to calculations of multiple dimorphism estimates confirms that Au. afarensis was only moderately dimorphic.

Introduction

Sexual dimorphism varies substantially in primates whether in response to sexual or ecological selection or a combination of both (Plavcan, 2012b). Gorillas and orangutans are highly dimorphic in canine size, body mass, and skeletal size, largely reflecting intense single male competition in both genera. In contrast, chimpanzees are moderately dimorphic with respect to mass and canine size and are essentially monomorphic in skeletal size. The latter principally reflects multimale group composition with patrilineal territorial defense and sperm competition (Morin, 1993; Reno et al., 2003). Modern humans show only moderate skeletal size dimorphism with low to moderate mass dimorphism (Reno et al., 2003). Moreover, in Homo sapiens both sexes exhibit dimorphic epigamic displays in facial hair, adipose distribution, and voice pitch (Lovejoy, 2009). Evaluating body mass dimorphism in humans is complicated by sex differences in body composition related to muscle and adipose mass (Plavcan, 2012b). Most striking, however, is the absence of human canine dimorphism brought about by dramatic feminization of the male tooth (Holloway, 1967; Washburn & Ciochon, 1974; Lovejoy, 2009; Suwa et al., 2009).

In recent years, dimorphism in early hominids has generated considerable discussion (Plavcan, 2002; Reno et al., 2003; Gordon, Green & Richmond, 2008; Reno et al., 2010; Plavcan, 2012b). It is now generally agreed that canine dimorphism is reduced in the earliest known hominids such as Sahelanthropus and Ardipithecus compared to other hominoids (Brunet et al., 2002; Suwa et al., 2009). On the basis of its relatively small canine, the ARA-VP-6/500 (‘Ardi’) skeleton is almost certainly female (Suwa et al., 2009), yet its postcranium suggests that the species exhibits fairly substantial body mass (50 kg) (Lovejoy et al., 2009). In combination with other individual postcranial specimens from Ardipithecus, it is likely that, as with chimpanzees and humans, skeletal size overlapped substantially in the two sexes.

Australopithecus afarensis, in which canine dimorphism is further reduced from its condition in Ardipithecus, is now represented by an extensive assemblage of specimens (Kimbel & Delezene, 2009). Initial discoveries of Au. afarensis included two associated partial skeletons each notable for its unusually small size. The first is A.L. 128-1 and A.L. 129-1 (A.L. 128/129). The second, A.L. 288-1 (“Lucy”), is more complete (Johanson & Taieb, 1976). In addition, the A.L. 333 site contains at least 9 adult individuals considered to be geologically contemporaneous (White & Johanson, 1989; Behrensmeyer, 2008). Pairwise comparisons of its largest individuals (e.g., A.L. 333-3) with Lucy have led to the common impression that size dimorphism was pronounced in Au. afarensis (Zihlman, Tobias & Coppens, 1976; McHenry, 1991; Richmond & Jungers, 1995). However, such comparisons entirely ignore specimens of intermediate size and the fact that Lucy and A.L. 333 could differ in age by tens of thousands of years (Johanson, Taieb & Coppens, 1982; Kimbel & Delezene, 2009), such that size variation may well reflect a variety of factors other than sexual dimorphism.

Over the past two decades, efforts have been aimed towards improving estimates of sexual dimorphism in fossils. Resampling procedures have been used to compare hominids with extant hominoids in order to better account for error in very small samples (Richmond & Jungers, 1995; Lockwood et al., 1996; Lee, 2001). While these procedures initially suggested that Au. afarensis was highly dimorphic, such inferences were limited to Lucy and larger specimens such as the A.L. 333-3 femur or the MAK-VP-1/3 humerus which may not represent ‘average’ females or males (Richmond & Jungers, 1995; Lockwood et al., 1996).

To address the problem of small sample size, techniques have been developed that simultaneously measure variation using multiple anatomical loci (Reno et al., 2003; Gordon, Green & Richmond, 2008). The Template Method (TM) can be used to calculate simple size ratios for multiple specimens that share an anatomical site with a more complete “template specimen” (e.g., Lucy) permitting the inclusion of many specimens not previously used in analyses of Au. afarensis dimorphism (Reno et al., 2003; Reno et al., 2010). When applied to this species the degree of dimorphism is similar to that of humans and intermediate between those of chimpanzees and gorillas.

Recently, Gordon and colleagues introduced the Geometric Mean Method (GMM) which cleverly takes advantage of the mathematical principle that the ratios of geometric means calculated across multiple variables are equal to the geometric mean of the ratios calculated for each individual variable (Gordon, Green & Richmond, 2008). Similar to the TM, the GMM enables estimation of dimorphism from a large multivariate sample. Surprisingly, when applied to Au. afarensis, it indicates that dimorphism was potentially greater than even that in gorillas and orangutans.

Gordon et al. proposed that this difference was attributable to error inherent in the TM resulting from: (1) biological error generated by using a template specimen to calculate size ratios; (2) allometric scaling relationships among variables, (particularly when the template specimen is a small individual); and (3) multiple representation of individuals from A.L. 333 (Gordon, Green & Richmond, 2008; Gordon, 2013). However, each of these concerns has been previously addressed. First, the TM has been evaluated through bootstrap simulations of chimpanzees, humans, and gorillas where the template individual is selected at random to account for intraspecific biological variation (Reno et al., 2003). This demonstrated that template size has no significant effect on the outcome of the procedure (Reno et al., 2010). Second, the effects of multiple representation of individuals from A.L. 333 was modeled by selecting randomly generated subsamples of reference hominoids to model the site’s minimum number of individuals (MNI) (Reno et al., 2003; Reno et al., 2005; Reno et al., 2010). Such modeling again demonstrated no significant effect on dimorphism estimates.

These findings suggest that the differences between the two methods must lie elsewhere. One possibility is differences in sample composition: forty-one specimens were used in the TM analysis whereas only 17 were used in the GMM analysis. A second potential factor is that the GMM allows more complete individuals such as Lucy and A.L. 128/129 to contribute multiple times to calculations of dimorphism. In addition, since these analyses were published, a new partial skeleton attributed to Au. afarensis, KSD-VP-1/1 or “Kadanuumuu,” has been described (Haile-Selassie et al., 2010). Despite the lack of cranial and dental elements for this specimen, the skeleton overlaps sufficiently with Lucy to warrant taxonomic assignment to Au. afarensis (Haile-Selassie et al., 2010). Based on a preliminary analysis, its estimated femoral head diameter (FHD) suggests that it is one of the largest specimens now known from Au. afarensis. Therefore, Kadanuumuu provides an opportunity to test the impact of a large bodied template specimen on estimates of Au. afarensis dimorphism, and to further test the contention that the TM is vulnerable to scaling effects. We also provide a direct comparison of the TM and GMM methods using identical samples that offers new insight into the methodological and sampling factors that underlie their divergent results.

Materials and Methods

Au. afarensis sample

Our current sample now includes 43 postcranial fossils spanning 16 different Afar localities (Table 1) (Lovejoy, Johanson & Coppens, 1982b; Lovejoy, Johanson & Coppens, 1982a; Harmon, 2006; Haile-Selassie et al., 2010; Plavcan, 2012b; Ward et al., 2012). A.L. 333 provides 26 skeletal elements sampled from approximately 9 individuals suggested by overlap of preserved dentitions (White & Johanson, 1989; Behrensmeyer, 2008) [but see (Plavcan et al., 2005) and below]. Fifteen remaining localities (i.e., the non-333 sample) span approximately 600,000 years (3.0–3.6 Ma) and provide 16 different individuals (the two specimens from A.L. 137 are both right humeri) (Johanson, Taieb & Coppens, 1982; Lovejoy, Johanson & Coppens, 1982b; Haile-Selassie et al., 2010; Ward et al., 2012). With the addition of Kadanuumuu, Lucy now enables the inclusion of 42 specimens in template calculations across 14 skeletal sites that can be used for estimating dimorphism in three overlapping samples: (1) the full Combined Afar (CA) that maximizes sample size, (2) A.L. 333 that avoids the influence of geographic and temporal size variation, and (3) non-333 which assures the sampling of only unique individuals. The A.L. 333 sample, however, is unchanged from our previous analysis so is not reanalyzed here (Reno et al., 2010). The Kadanuumuu template sample includes only 7 A.L. 333 and 7 non-333 specimens measured across 5 skeletal sites. Separately these samples are too small to produce reliable dimorphism estimates, so only the CA analysis is explored for Kadanuumuu.

Table 1 Au. afarensis sample used in the template method simulations.

Metrics	Specimens	A.L. 288-1 template estimated FHD with	KSD-VP-1/1 template estimated DSTB	
CLAV: Max. mid-shaft diameter of the clavicle	A.L. 333x-6/9		18.7	
HHD: Max. humeral head diameter	A.L. 333-107a	39.4		
HOCB: ML width of humerus measured tangent to the superior margin of the olecranon fossa	A.L. 137-48A	32.6		
A.L. 137-50	38.3		
A.L. 223-23	35.3		
A.L. 333-29	33.2		
A.L. 333w-31	34.3		
MAK-V/P-1/3	37.8		
HARB: ML breadth of the distal articular surface	A.L. 137-48A		20.9	
A.L. 322-1		19.7	
CAPD: Max. diameter of capitulum	A.L. 322-1	32.2		
A.L. 333w-22	39.5		
A.L. 444-14	37.2		
RHD: Max. diameter of the radial head	A.L. 333x-14bc	44.3		
A.L. 333x-15bc	44.5		
ULB: ML width of ulna immediately distal to radial facet	A.L. 333x-5	37.1		
A.L. 333w-36	29.8		
A.L. 438-1a	40.9		
FHD: Max. femoral head diameter	A.L. 152-2	33.1		
A.L. 288-1ap	28.6		
A.L. 333-3	40.9		
A.L. 827	38.1		
FNKH: Femoral neck height normal to long axis at midpoint	A.L. 333-117	38.7		
A.L. 333-123	33.0		
A.L. 333-142d	30.1		
TRCD: Max. femoral shaft diameter immediately below lesser trochanter	A.L. 211-1	36.4		
A.L. 333-95bc	35.3		
MAK-V/P-1/1	34.4		
GSTB: AP femoral width immediately above gastrocnemius tubercle	A.L. 333-4	35.2		
A.L. 333w-56	33.6		
A.L. 333-140d	30.2		
FLCL: AP length of the lateral femoral condyle	A.L. 129-1a		19.0	
A.L. 333-4		23.4	
A.L. 333w-56		23.6	
CNDC: ML distance between centers of medial and lateral tibial condyles	A.L. 129-1b	27.9		
A.L. 333x-26	38.5		
A.L. 333-42	36.7		
PXTB: Max. ML proximal tibial breadth	A.L. 330-6	37.4		
TMXT: Max. AP breadth at tibial tuberosity	A.L. 330-6		22.2	
A.L. 333x-26		22.5	
DSTB: AP tibial distal articular breadth	A.L. 288-1		17.2	
A.L. 333-6	37.2	22.4	
A.L. 333-7	42.9	25.8	
A.L. 333-96	38.4	23.1	
A.L. 545-3	31.9	19.2	
KSD-VP-1/1	44.6	26.8	
FIBD: Max. diameter of distal fibula	A.L. 333-9A	42.8		
A.L. 333-9B	38.9		
A.L. 333w-37	37.8		
A.L. 333-85	40.6		
TAL: Max. AP length of talus	A.L. 333-147	36.0		
Notes.

a Because of slight eccentricity in this specimen the average of the ML and AP diameters was used instead.

b Possible antimeres.

c Inclusion of these large bodied subadult specimens increases Au. afarensis dimorphism value.

d These small bodied specimens lacking epiphyseal fusion are larger than the smallest adult A.L. 333 specimen. Exclusion or subsequent growth would reduce Au. afarensis value.

Dimorphism estimation

Template method

While large samples exist of Au. afarensis postcranial fossils, relatively few represent the same anatomical loci. As such, much of the variation in the sample reflects size differences between anatomical parts (Fig. 1A). Fortunately, the existence of a relatively complete skeleton can serve as a template to account for the differences in size between these sites. To apply the template method, the metric from each fossil is converted to a simple size ratio by dividing each by the homologous metric from a template specimen (Lucy or Kadanuumuu) (Fig. 1B). This calculation preserves the relative dispersion of the individuals within each variable and enables these dimensionless size ratios to be compared across variables. To standardize size ratios and to facilitate inspection of the data, we multiplied each by a reference metric commonly used in size estimation: femoral head diameter (FHD) for the Lucy template and distal tibia breadth (DSTB) for Kadanuumuu (Table 1). This step has no effect on the estimate of dimorphism because it is simply multiplying each ratio by a constant leaving the dispersion of the sample unchanged (Fig. 1C). The sample is then used to compute the Coefficient of Variation (CV) and Binomial Dimorphism Index (BDI) as in previous analyses (Lovejoy et al., 1989; Reno et al., 2003) with each specimen providing an equal contribution to the dimorphism estimate.

Figure 1 Demonstration of the calculation for the template method.

(A) Many Au. afarensis specimens can be used to judge the species’ skeletal size distribution, but only a few represent the same skeletal site (horizontal axis) and therefore cannot be simply compared with one another. However, many sites are also found in the Lucy skeleton (diamonds). (B) The relative size of each non-Lucy specimen can therefore be calculated as a simple ratio (vertical axis) of each specimen to the same site as preserved in Lucy. These (dimensionless) ratios can then be used to compute a CV or BDI for the species. Note that the template ratios all equal 1 for the template specimen, so the choice of metric used for this individual has no effect on the dimorphism calculation. (C) To convert ratios to “real dimensions” each can be “normalized” using Lucy’s FHD. This has no effect on the sample’s dispersion (i.e., compare B to C), and therefore has no impact on the value of the CV and BDI when Lucy (or other specimen) is used as a “template.” See Table 1 for variable definitions.

Because the template specimen serves as the denominator for each template ratio, it produces a value of “1” for its own corresponding metric in each variable (gray diamonds in Figs. 1B and 1C). As such, it makes no difference which variable is chosen to represent the template specimen. However, to ensure equal representation of each of the other individuals, we used only one metric in cases where multiple variables are measurable. We followed the convention of opting first for an articular dimension, if available, followed by a diaphyseal shaft diameter (i.e., for CA Lucy template sample: A.L. 128/129, distance between centers of tibial condyles, CNDC; A.L. 322-1, humeral capitulum diameter, CAPD; A.L. 333-3, FHD; MAK-V/P-1/1, maximum diameter below the lesser trochanter, TRCD). This was based on the tendency for such variables to have a greater association with size (Gordon, Green & Richmond, 2008). This selection convention had little effect on dimorphism estimates. The range of CVs from each of the possible combinations of the available metrics from these four specimens in the CA sample was 11.36–12.16 with a mean of 11.72. This compares to the CV of 11.75 for the sample used in this analysis. Simulations for some of the samples using different metrics show no meaningful effect on the results.

Geometric Mean Method

The Geometric Mean Ratio (GMR) is calculated as follows: (1) A mean value is calculated for each metric and sex is assigned to each individual (values above the mean are presumed to be male; those below the mean are presumed to be female). (2) Dimorphism ratios are calculated for each variable using the geometric mean of presumed males divided by the geometric mean of presumed females. (3) The GMR is calculated as the geometric mean of the dimorphism ratios of all of the separate metric ratios (Gordon, Green & Richmond, 2008). While not relying on a template, the GMM does require at least two specimens to represent each anatomical site. In addition, in its previous application the GMM includes all available metrics of more complete specimens, such as Lucy, Kadanuumuu and the others mentioned above, in the calculation of each skeletal site (Gordon, Green & Richmond, 2008). We have applied the GMM to both the full CA Lucy (Table S1) and Kadanuumuu (Table S2) template samples.

Sampling and bootstrapping procedures

As the Au. afarensis BDI, CV and GMR statistics are computed from a combination of different variables which cannot be assumed to have the same variance, they cannot simply be compared to similar statistics computed using a more standard metric (e.g., actual FHD). Therefore, we conducted simulations of randomly generated samples with identical anatomical composition from species with known sexual dimorphism (Fig. 2 and Table S3). The chimpanzee, human, and gorilla samples used here are similar to those used previously (Reno et al., 2003; Reno et al., 2010) with the addition of three new metrics to maximize use of the Kadanuumuu template (Table 1). These three hominoids are appropriate reference taxa with which to judge Au. afarensis dimorphism, because they represent three of the four extant species most closely related to early hominids. In addition, with the exception of some ceropithecoids (i.e., Mandrillus (Setchell et al., 2001)) and possibly the extinct Miocene genus Lufengpithecus as indicated by postcanine dentition (Kelley & Xu, 1991) they essentially encompass the range of primate skeletal dimorphism (Smith & Jungers, 1997). Therefore, to determine where Au. afarensis falls within the spectrum of African hominoid dimorphism, we tested the hypotheses that its dimorphism is above minimally dimorphic chimpanzees, below extremely dimorphic gorillas, or compatible with moderately dimorphic humans.

Figure 2 Sampling procedure used to simulate the Template and Geometric Mean Methods in extant humans, chimpanzees and gorillas.

For the TM, we used a sample of metrics that mirror the anatomical composition of each respective Au. afarensis assemblage (Fig. 2). A template specimen was chosen randomly for each individual sample. The ratio between each metric and its homologue in the template was computed for each specimen. These rescaled metrics were used to calculate the BDI and CV. This sampling procedure was repeated 1,000 times for each species as in previous analyses (Lockwood et al., 1996; Reno et al., 2003; Harmon, 2006; Reno et al., 2010). These simulations demonstrate that the TM accurately reflects the relative dimorphism of chimpanzees, humans and gorillas, suggesting that biological error from the template specimen is small relative to variation in skeletal dimorphism (Reno et al., 2003). Similarly, for the GMM, random samples with the same anatomical composition as the fossil assemblage were selected for computation of the GMR. As in the previous analysis of Gordon, Green & Richmond (2008) the metrics corresponding to those more complete specimens such as Lucy, A.L. 128/129, and Kadanuumuu (and a few others) were each sampled from single individuals.

Specimens sourced from non-333 localities obviously represent separate individuals, so each metric (TM) or individual (GMM) is randomly sampled from the full comparative sample with replacement. However, specimens from A.L. 333 may represent co-mingled individuals. Based on dentitions, at least 9 individuals are preserved at this site. Therefore, to model the A.L. 333 depositional event, a subsample of 9 individuals was randomly selected without replacement for each iteration (Fig. 2). These served as the source of metrics with replacement to represent the A.L. 333 assemblage. This often produces samples based on fewer than the full 9 individuals as may have happened in the accumulation of the A.L. 333 postcranial assemblage. This approach most accurately recreates the known composition and current understanding of the deposition of A.L. 333 assemblage.

Alternatively, the MNI from A.L. 333 is strictly only three based on the occurrence of three left distal fibulas. However, our original sample also includes a decidedly smaller individual (A.L. 333w-36 ulna) and at least one and possibly two large subadult individuals (A.L. 333x-14 and A.L. 333x-15 unfused radial heads and A.L. 333-95 proximal femur (Reno et al., 2003)). We have previously conducted simulations sampling as few as 5 individuals representing this remotely possible sampling at A.L. 333. These smaller tests did not substantively alter our results (Reno et al., 2005). Our current sample now includes at least one additional smaller subadult specimen (A.L 333-140 and A.L. 333-142 distal and proximal femoral fragments) (Ward et al., 2012). This raises the MNI to 6. However, as an MNI of 5 was also used in the previous analysis by Gordon, Green & Richmond (2008), we explored how such a restriction would impact some of our simulations. Randomly sampling from as few as 5 individuals frequently generates samples composed of only a single sex, which are obviously not appropriate for evaluating sexual dimorphism. Therefore, we required that each subsample of 5 individual include at least one member of each sex. All simulations were programmed in Fortran.

Results

Kadanuumuu does not alter the Au. afarensis dimorphism prediction

The inclusion of Kadanuumuu dramatically increases the upper size boundary of the non-333 assemblage, which now surpasses the entire range of A.L. 333 (Fig. 3). For the full CA sample the value of Au. afarensis dimorphism falls well within the human distribution of simulated values and only minimally overlaps with chimpanzees or gorillas (Fig. 4A; for BDI results see Fig. S1). In particular, the A. afarensis CA sample is significantly lower (one-tailed) than that of gorillas (Table 2). Results are similar when the A.L. 333 fossils are modeled as only representing 5 individuals (Table S4). The smaller non-333 sample also shows a similar pattern (Fig. 4B), although it just fails to reach statistical significance at p = 0.05 level for both chimpanzee (low dimorphism) and gorillas (high dimorphism) (Table 2). These data confirm previous analyses using the TM that indicate that Au. afarensis skeletal dimorphism was significantly below that of gorillas and unlikely to be as low as chimpanzees. The possibility of dimorphism similar to humans could not be rejected.

Figure 3 Size distribution of fossils using Lucy as a template.

Each specimen’s assigned sex and the number of times it is included in the respective GMM analyses are indicated. (A) The addition of Kadanuumuu elevates the non-333 range to greater than that of A.L. 333. Note that Lucy and A.L. 128/129 account for 10 of 11 female assignments in the Gordon, Green & Richmond (2008) analysis. (B) In the full, unmodified CA sample these two specimens account for 18 of 30 female assignments resulting in minimal overlap between sexes. (C) When Lucy values are increased to 130% her original size or (D) when Lucy and A.L. 128/129 are allowed to contribute only once to the sample, the sex assignments overlap substantially.

Figure 4 Frequency histograms of the simulations modeling the Template Method using extant chimpanzee, human and gorilla reference samples (1,000 iterations each).

The vertical line and number indicate the dimorphism value (CV) for the respective Au. afarensis sample using either Lucy (A, B, and D) or Kadanuumuu (C) as a template. In all cases the Au. afarensis value is compatible with human levels of dimorphism. However, as sample sizes decrease (B–D), the loss of intermediate specimens in the Au. afarensis sample increases the value of hominid dimorphism and/or the range of generated dimorphism values for each of the reference taxa. In these cases, the Au. afarensis values fall in the overlap between humans and gorillas demonstrating the importance of maximizing sample size when estimating dimorphism.

Table 2 Descriptive statistics and exact counts from Template and Geometric Mean Method simulations.

Simulation	Dimorphism value	Chimp	Human	Gorilla	
		Mean (sd)	<	>	Mean (sd)	<	>	Mean (sd)	<	>	
Lucy template—CA (N = 42)											
BDI	1.207	1.162 (0.025)	959	41	1.205 (0.032)	539	461	1.290 (0.037)	11	989	
CV	11.75	9.53 (1.33)	922	78	11.69 (1.58)	537	463	15.40 (1.52)	8	992	
Lucy template—non-333 (N = 16)											
BDI	1.214	1.164 (0.035)	920	80	1.197 (0.041)	690	310	1.287 (0.051)	79	921	
CV	12.47	9.79 (1.91)	916	84	11.51 (2.14)	701	299	15.77 (2.32)	73	927	
Kadanuumuu template—CA (N = 14)											
BDI	1.234	1.152 (0.39)	970	30	1.206 (0.053)	723	277	1.265 (0.051)	281	719	
CV	13.20	9.14 (2.07)	948	52	12.25 (2.83)	653	347	14.85 (2.43)	245	755	
Lucy template—CA (N = 13)											
BDI	1.255	1.157 (0.039)	989	11	1.201 (0.050)	851	149	1.266 (0.057)	426	574	
CV	14.44	9.47 (2.13)	986	14	11.84 (2.53)	845	155	14.97 (2.63)	426	574	
Geometric Mean Method											
Full Lucy sample	1.299	1.118 (0.018)	1,000	0	1.157 (0.025)	1,000	0	1.246 (0.040)	900	100	
130% Lucy	1.179	1.118 (0.018)	999	1	1.157 (0.025)	809	191	1.246 (0.040)	33	967	
Single Lucy & 128/129	1.176	1.117 (0.019)	998	2	1.156 (0.025)	781	219	1.241 (0.039)	35	965	
KSD sample	1.382	1.115 (0.028)	1,000	0	1.158 (0.040)	1000	0	1.247 (0.056)	988	12	
Notes.

< number of iterations that fell below hominid value

> number of iterations that fell above hominid value

Template size does not affect dimorphism estimates

Our previous experiments have demonstrated that there is no association between template size and projected dimorphism in chimpanzees, humans and gorillas and there are therefore no significant allometric effects inherent in template size (Reno et al., 2010). The large size of Kadanuumuu can now be used to test the hypothesis that the TM when applied to Au. afarensis is also free of allometric effects. The template sample available for Kadanuumuu is smaller than that for Lucy. However, as these two specimens and A.L. 128/129 are all included in both samples, the same total size range is maintained while many intermediate specimens are lost (Fig. 5).

Figure 5 Estimated distal tibial articular breadth using Kadanuumuu as a template and sex assignment from the GMM.

This smaller sample includes the full Au. afarensis size range from Lucy to Kadanuumuu but excludes numerous intermediate sized specimens available when Lucy is used as template. Note that Kadanuumuu, Lucy and A.L. 128/129 account for 10 of 21 measurements included in the calculation of the GMR.

When compared to the simulations, the smaller Kadanuumuu template sample does predict greater dimorphism than does the full CA Lucy template sample (Fig. 4C). However, is this effect the product of the size of Kadanuumuu or the resulting reduction in sample size? To examine this question we restricted the Lucy template sample to the same specimens used in the Kadanuumuu analysis (except for the A.L. 333-6/9 clavicle which does not overlap with Lucy). This produced a sample of 13 total specimens, whose CV and BDI are also greater than the full 42 specimen sample (Fig. 4; Table 2). The dimorphism of the smaller Lucy sample also fell in the overlap between humans and gorillas (Fig. 4D). In fact, it predicted a slightly higher level of dimorphism than did the Kadanuumuu template. The TM method thus performed essentially the same with both very small and very large templates, illustrating again that template size has no effect on sample variation and inferred dimorphism. However, reduction in sample size and particularly the loss of intermediate sized specimens does greatly impact sample dimorphism.

The use of template ratios has no significant effect on dimorphism estimates

The strikingly different results obtained with the TM versus those generated by the GMM is puzzling (Reno et al., 2003; Gordon, Green & Richmond, 2008; Reno et al., 2010). To search for its underlying cause, we performed a direct comparison of the GMM on our Au. afarensis samples. As Table 3 shows, GMM results are identical whether raw metrics or template ratios are used for the calculation, which, as we have previously demonstrated has no effect on the TM as well (Reno et al., 2005; Reno et al., 2010). This can also be confirmed through inspection of Fig. 1. While the GMM is normally applied to the raw metrics depicted in Fig. 1A, the generation of the template ratios preserves the relative size between specimens within variables. Thus, the GMR computed on the size ratios or even the estimated FHD will be exactly the same. Therefore, the use of a template specimen does not introduce any directional scaling effects between variables, and the TM and GMM should theoretically perform similarly given identical samples.

Table 3 Computation of the Geometric Mean Ratio (GMR) for two variables.

GMRs are identical whether raw metrics or template ratios are used.

Specimen	HOCB	HOCB ratio	Specimen	FNKH	FNKH ratio	
A.L. 322-1	31.7	0.975	A.L. 288-1	21.8	1.000	
A.L. 288-1	32.5	1.000	A.L. 333-142	22.9	1.050	
A.L. 137-48A	37.1	1.142	MAK-VP-1/1	23.8	1.092	
A.L. 333-29	37.7	1.160	A.L. 333-123	25.1	1.151	
A.L. 333w-31	39.0	1.200	A.L. 333-95	26.0	1.193	
A.L. 223-23	40.1	1.234	A.L. 333-3	29.1	1.335	
MAK-VP-1/3	43.0	1.323	A.L. 333-117	29.5	1.353	
A.L. 137-50	43.5	1.338				
Mean	38.075	1.172		25.457	1.168	
Female geo mean	34.647	1.066		23.369	1.071	
Male geo mean	41.356	1.272		28.155	1.291	
GMR	1.194	1.194		1.205	1.205	
Notes.

HOCB ML width of humerus measured tangent to the superior margin of the olecranon fossa

FNKH Femoral neck height normal to long axis at midpoint

The Geometric Mean Method still predicts great dimorphism in an expanded Au. afarensis sample

If not template ratios or scaling effects, then perhaps differences in sample composition underlie the different results from the two methods. Gordon, Green & Richmond (2008) used a sample of 17 fossils. As the MNI at A.L. 333 was nine, this must represent approximately 14 unique individuals. Our sample included over 40 fossil specimens, likely representing 25 separate individuals. We also used different methods to account for A.L. 333’s taphonomic issues. Gordon, Green & Richmond (2008) repeatedly randomly assigned the 12 metrics representing A.L. 333 specimens to 5 composite individuals for each iteration in their simulations. The total fossil assemblage was then modeled as representing 10 individuals (5 for the non-A.L. 333 localities and 5 for the composite A.L. 333 individuals).

To directly compare the two methods, we applied the GMM to the same sample we used in the TM above (Table S1). We accounted for A.L. 333 by drawing its contribution to the sample from either 9 randomly chosen individuals or from 5 (mixed-sex—see above) individuals (Reno et al., 2010). Remarkably, the GMR was nearly identical in both our and (Gordon, Green & Richmond, 2008)’s samples (1.299 versus 1.291). This value fell within the upper range of the gorilla simulation (top 10%) and did not overlap with those of either chimpanzees or humans (Fig. 6A, Table 2). Thus, neither the different sample compositions nor modeling methods can account for our dramatically different results.

Figure 6 Frequency histograms of the simulations modeling the Geometric Mean Method using extant chimpanzee, human and gorilla reference samples (1,000 iterations each).

The vertical line and number indicate the dimorphism value (GMR) for the respective Au. afarensis sample and the gray line in (A) represents GMR if Lucy is increased to 130% original size. In (B) Lucy and A.L. 128/129 contribute only femoral head diameter or proximal tibial breadth respectively. When small and more complete specimens such as Lucy and A.L. 128/129 contribute multiple metrics, predicted dimorphism values fall in the upper range of the gorilla distribution (A and C). However, when Lucy is scaled to an intermediate size or Lucy and A.L. 128/129 are restricted to contributing a single metric, the Au. afarensis falls in the human distribution and significantly outside those of the apes (A and B).

Multiple inclusion of more complete specimens augments dimorphism in the GMM

A final difference between TM and the GMM is that more complete specimens, such as Lucy, A.L. 128/129, and Kadanuumuu, are permitted to contribute multiple metrics in the GMM, but not in the TM. In the Gordon, Green & Richmond (2008) sample, Lucy and A.L. 128/129 contributed 10 of 26 metrics from the 17 specimens (Fig. 3A). What was the effect of this repeated contribution by a few specimens which lie at the margin of the size range?

One way to explore this is to examine the effect of Lucy’s small size directly. The Kadanuumuu DSTB is 156% that of Lucy, and lies at the upper limit of the size range of A. afarensis. We systematically increased all of Lucy’s metrics in 10% increments and recomputed the GMR using Gordon, Green & Richmond’s (2008) methods. Increasing the size of Lucy to 130% of its original size (approximately half way to Kadanuumuu) reduced the GMR for the entire sample to 1.179 from the original value of 1.299. This placed Au. afarensis within the human range and significantly below that of gorillas (Fig. 6A, Table 2). At 160% original size, the sample GMR returned to only 1.242.

For additional testing we also modified the size of the small ulna, A.L. 333w-36, to determine how this particular specimen impacts dimorphism. As it is slightly larger than Lucy we modified its size in 10% increments from 90% to 160% original size. The largest changes were a reduction to 1.294 at 110% original size and an increase to 1.307 for 160% original size. Thus, the random co-occurrence of Lucy’s small size and substantial preservation has a disproportionate effect on the calculation of the GMR and inferred value of Au. afarensis dimorphism.

To explore this effect on calculation of the GMR by another route, we did not change Lucy’s size but restricted her contribution to only a single metric. This required the elimination of three variables, humeral head diameter (HHD), talus length (TAL) and radial head diameter (RHD), because absent Lucy there were no other pairings for these skeletal sites. Allowing Lucy to contribute only a single metric reduces the GMR to a range of 1.217 (distance between tibial condyle centers, CNDC) to 1.247 (maximum diameter of distal fibula, FIBD) depending on the metric chosen to represent Lucy. However, the similarly diminutive A.L. 128/129 also contributes 4 metrics to the sample. We thus calculated all possible GMRs where both Lucy and A.L. 128/129 each contributed only a single metric. This still allowed the GMR to be computed on 10 or 11 variables containing at least 37 fossils. Such reduction resulted in a GMR ranging from 1.154 (both specimens contributing femoral shaft diameter at gastrocnemius insertion) to 1.216 (with Lucy contributing FIBD and A.L. 128/129 contributing CNDC) with an average of 1.179. To determine the significance of this change, we simulated the effect of including only Lucy’s FHD and A.L. 128/129’s PXTB. In this case, the calculated GMR (1.176) fell within the human distribution (Fig. 6B, Table 2) and was significantly different from gorillas and chimpanzees (or nearly so when restricting the A.L. 333 MNI to 5, Table S4); a result that is very similar to that obtained using the TM.

The effect of multiple inclusion is further demonstrated by applying the GMM to the smaller Kadanuumuu template sample where A.L. 128/129, Lucy and Kadanuumuu account for 10 of the 21 metrics. This resulted in an estimated GMR (1.383) that falls at the upper margin (top 2%) of the gorilla distribution and suggests that Au. afarensis would be significantly more dimorphic than one of the most dimorphic living primates (Fig. 6C, Table 2). This is unlikely as the ratio between even the most extreme known Au. afarensis specimens (i.e., the absolutely largest and smallest specimens) does not surpass those of gorillas (see below). Thus, the repeated contribution of a few extreme sized individuals has a dramatic impact on the perceived variation of the sample.

Discussion

Similarities in the template and Geometric Mean Methods

The primary strength of both the TM and GMR is their greatly expanded sample size. When sex is unknown, a large sample is the only means to ensure that both sexes are adequately represented and that a full distribution of the species is appropriately included (Simpson, Roe & Lewontin, 1960; Koscinski & Pietraszewski, 2004). The importance of maximizing sample size is demonstrated by the results of this analysis. For the largest CA sample using Lucy as a template, Au. afarensis is significantly different from both chimpanzees (using the BDI) and gorillas and is indistinguishable from human levels of dimorphism (Table 2). Similar results were obtained with the smaller non-333 sample and previously with the A.L. 333 sample (Reno et al., 2010). The failure of these smaller samples to obtain significance (although still with a low probability of compatibility) from chimpanzees and gorillas at either extreme of the dimorphism scale results from the increased variance for each species in simulated distributions (Table 2). The cause of such variability is demonstrated by the Kadanuumuu template analysis, as smaller sample sizes may fail to be representative of the species’ distributions (Fig. 5). Gordon, Green & Richmond (2008) used a smaller sample because they focused their analysis on variables shown to scale isometrically with body mass across primates. This was motivated by the fact that mass dimorphism has played a prominent role in analyses of primate ecology, reproductive behavior, and sexual selection. However, mass and skeletal dimorphism do not have the same relationships in humans and apes, making the task of inferring mass dimorphism from skeletal dimorphism difficult or impossible for early hominids (Reno et al., 2003; Gordon, Green & Richmond, 2008; Plavcan, 2012b). We have avoided this conundrum by simply focusing on skeletal dimorphism, an important indicator of total size and sex differences in growth, which are two central factors in primate reproductive biology (Hamada & Udono, 2002; Reno et al., 2003).

A resolution of the contradiction in results obtained by the template and Geometric Mean Methods

As demonstrated here, the discrepancy between results generated by the TM and GMM stems from the fact that more complete specimens such as Lucy, A.L. 128/129, and Kadanuumuu that sample the margins of the size range contribute multiple metrics to the calculation of the GMR. In the original application of the GMM, the metrics contributed by Lucy and A.L. 128/128 accounted for 10 of 11 metrics that fell below the means of each of the respective variables in the sample (Gordon, Green & Richmond, 2008). Gordon, Green & Richmond (2008) accounted for this by assigning the smallest two individuals from the 10 (one individual for each of the 5 separate localities and 5 for A.L. 333) resampled humans or apes to represent the multiple metrics derived from Lucy or A.L. 128/129. This did impact the generated dimorphism values from the extant hominoids, but not sufficiently to match the dimorphism observed via the TM. This may be because constraining the two smallest individuals of 10 sampled hominoids does not adequately account for the fact that not only are Lucy and A.L. 128/129 the smallest adult postcranial Au. afarensis specimens known from among the approximate 25 individuals in our postcranial samples, but Lucy is also one of the smallest two or three specimens within a larger pool of mandibular and dental dimensions (Lockwood, Kimbel & Johanson, 2000; Haile-Selassie & Melillo, 2014).

The larger sample used here permitted the direct exploration of the effect that multiple representation of these small individuals had on inferring dimorphism. Lucy and A.L. 128/129 still account for 18 of 30 metrics assigned female. This produces a situation similar to earlier analyses that were driven by pairwise comparisons between Lucy and a few larger specimens. As such, Lucy and A.L. 128/129 still dominate the contributions of inferred females in the individual dimorphism ratios. This results in little overlap between the inferred sexes in the calculation of the GMR (Fig. 3B). Limiting their contributions to single metrics or adjusting the size of Lucy brought the GMR well within the human range and significantly outside those of gorillas and chimpanzees. This is because there is greater overlap between the inferred sexes of the individual fossils (Figs. 3C and 3D).

Lucy and the typical Au. afarensis female

These differences in the distributions of inferred sexes raise the question of which dimorphism pattern best characterizes Au. afarensis? A number of lines of evidence support a more human-like pattern with substantial overlap between the sexes. The first is the agreement between the TM and the GMM when all specimens are allowed to contribute only once to the dimorphism estimate with modern human levels of dimorphism. In these cases, A.L. 128/129 and Lucy are weighted equally to all other specimens and do not overly influence female size.

A second is that many of the size ratios between extreme specimens such as Lucy, A.L. 128/129, Kadanuumuu, and A.L. 333-3 can be accommodated within human distributions. When compared to sample ranges, the difference between Lucy and A.L. 333-3 FHD (1.43) is matched or surpassed in our samples of 50 humans or gorillas (Table 4). In addition, the difference in DSTB of Lucy and Kadanuumuu (1.56) is also found within the range of the human and gorilla comparative samples (Table 4). Furthermore, all maximum size differences between Au. afarensis specimens can be accommodated within the ranges of each gorilla variable. This suggests that Au. afarensis is not more dimorphic than gorillas, and results indicating this emerge from the over emphasis of unusually complete small individuals.

Table 4 Maximum/minimum ratios femoral head diameter (FHD) and distal tibial breadth (DSTB) in the extant hominoid and Au. afarensis samples.

Species	FHD	DSTB	
Gorilla	1.559	1.741	
Human	1.431	1.697	
Chimpanzee	1.287	1.419	
A.L. 333-Lucy	1.430	–	
Kadanuumuu-Lucy	–	1.558	

Lastly, the three most closely related living taxa to Au. afarensis, humans, chimpanzees, and bonobos, all show substantial overlap between the sexes. There was also likely overlap between the sexes in Ardipithecus and potentially South African gracile hominids Au. africanus and Au. sediba (Harmon, 2009; Lovejoy et al., 2009; Berger et al., 2010). While early Homo displays substantial morphological diversity, there is little systematic evidence to infer strong dimorphism in these taxa (Ruff, 2002; Plavcan, 2012a).

Appreciable overlap between the sexes would suggest an absence of targeted selection for increased male size. The female Ardi skeleton is estimated to have a body size (50 kg), which is similar to those of the largest Au. afarensis specimens (e.g., 50-70 kg for A.L. 333-3 depending on prediction method) (McHenry, 1992), suggesting that early hominid size was generally stable. Given Ardi, both Plavcan (2012b) and Gordon (2013) have argued that small individuals such as Lucy, A.L. 128/129 and STS-14 (Au. africanus) imply that any dimorphism increase in Australopithecus likely resulted from a reduction in average female size. Such size reduction might reflect selection associated with earlier reproduction, reduced energy and resource utilization, and increased fecundity (Lovejoy, 2009; Gordon, 2013). This would accord with the expanded demographic success of australopithecines compared to extant hominoids (Reno, 2014). Alternatively, the presence of these diminutive and presumably female specimens may simply reflect species variability yet to be observed in the time restricted Ar. ramidus sample (Suwa et al., 2009).

The relatively stable size patterns observed between Ardipithecus and Australopithecus suggest there was not strong selection for greater male body size that would result from a reproductive strategy arising from increased individual male reproductive success via inter-individual aggression. In fact, the reduction in canine dimorphism with feminization in the male would argue for reduced “agonistic” behaviors (Lovejoy, 2009). This is particularly so given the strong association between canine dimorphism and reproductive behavior in anthropoids (Plavcan, 2012b) and the lack of a dramatic dietary shift associated with canine modification in early hominids (Suwa et al., 2009).

Supplemental Information

Figure S1 Frequency histograms of the simulations modeling the Template Method using extant chimpanzee, human and gorilla reference samples (1,000 iterations each)

The vertical line and number indicate the dimorphism value (BDI) for the respective Au. afarensis sample. The BDI produces essentially similar results as the CV depicted in Fig. 3.

Click here for additional data file.

Table S1 The Lucy (A.L. 288-1) template sample used to calculate Geometric Mean Method dimorphism

Bold indicates the values used for the simulations limiting A.L. 128/129 and Lucy contribution to a single metric. The values for Lucy increased by 130% (see text) are shown in italics.

Click here for additional data file.

Table S2 The Kadanuumuu (KSD-V/P-1/1) template sample used to calculate Geometric Mean Method dimorphism

Click here for additional data file.

Table S3 Descriptive statistics for individual metrics measured directly from chimpanzee, human and gorilla specimens

Click here for additional data file.

Table S4 Results of simulations modeling only 5 mixed sex individuals preserved at A.L. 333

Click here for additional data file.

We thank Yohannes Haile-Selassie, Curator of Physical Anthropology at the Cleveland Museum of Natural History, for access to comparative specimens in his care and Lyman Jellema for curatorial assistance. We also thank William Brouwer of Penn State Research Computing and Cyberinfrastructure (RCC) for his assistance.

Additional Information and Declarations

Competing Interests

Author Contributions

The authors declare there are no competing interests.

Philip L. Reno conceived and designed the experiments, performed the experiments, analyzed the data, contributed reagents/materials/analysis tools, wrote the paper, prepared figures and/or tables.

C. Owen Lovejoy conceived and designed the experiments, reviewed drafts of the paper.

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
