# Peer review of "From Lucy to Kadanuumuu: balanced analyses of Australopithecus afarensis assemblages confirm only moderate skeletal dimorphism"

_PeerJ, doi:10.7717/peerj.925_

## Round 0.1 · original submission · Major Revisions

· Academic Editor

Major Revisions

In view of the long history of debate over the assumptions, methods and inferences presented here, several reviewers were consulted -- including one of the 2 that the authors wish to avoid (Gordon). The reviews range from accept (n=1) to minor revision (n=1) to major revision (n=2). A 5th review was received but did not make it into the PEERJ folder, and I include that detailed review in quotes below:

“This paper revisits the contested issue of the level of size dimorphism in Australopithecus afarensis by repeating a previous Template Method analysis with the addition of more recently discovered specimens from Hadar and Woranso-Mille, and using the Woranso-Mille partial skeleton, KSD-VP-1/1, as an alternative template specimen to AL288-1. Whether using AL288-1 or KSD-VP-1/1, the template method results show that the A. afarensis sample is not significantly different than dimorphism in humans, is significantly lower than gorillas in one of four analyses, and is significantly higher than chimpanzees in all but one scenario. The study also simulates the Geometric Mean Method under several scenarios. The study concludes that, when correcting for species completeness, and when omitting analyses that under-sample fossils of intermediate size, A. afarensis was moderately dimorphic.
New fossils warrant reanalysis of long-standing debates such as this one. However, the Template Method has been widely criticized on multiple grounds, by multiple research groups. This manuscript cannot simply ignore these issues. Therefore, it either needs to be replaced by methods that avoid its assumptions, and perform better with known datasets, or improved and tested on known datasets, before it can be applied to the fossil record. These problems, together with a number of major issues outlined below, prevent me from being able to recommend that this article be accepted.
Major issues:
The “Template Method” has been shown to be highly problematic (Plavcan et al, 2005; Scott & Stroik, 2006; Gordon et al 2008; Gordon 2013). This manuscript does not adequately address those issues.
Contrary to the authors’ claims (page 6), the large size of KSD-VP-1/1 (Kadanuumuu) cannot be used to test the contention that the Template Method is subject to allometric effects. First, KSD-VP-1/1 is not known to belong to the same species as AL 288-1. If it is the same species, it is not an adequate test because two individuals are insufficient to demonstrate the allometric relationships among anatomical regions. The authors need to demonstrate how the Template Method is affected by violations of the assumptions of isometry, and the assumption that the intercept passes through the origin. Violation of either of these assumptions results in ratios that change with size, in contrast to the Template Method’s assumption that the ratio does not change with size. These problems and others have been discussed in several articles (Plavcan et al, 2005; Scott & Stroik, 2006; Gordon et al 2008; Gordon 2013). This manuscript cannot simply ignore these issues.
An example of why the Template Method is problematic is seen in the authors’ own data. The Template Method produces different results depending on the measurement selected. For example, AL 128-1/129-1 has two preserved measurements in common with KSD-VP-1/1. The Template Method yields different estimates for the distal tibia size (DSTB) depending on what other anatomical measure is selected:
DSTB = 16.4, if based on TMXT
DSTB = 19.0, if based on FLCL
This illustrates how different skeletal elements will have different relationships with each other, with several important implications:
1. The Template Method makes major assumptions, and these assumptions are violated.
2. The Template Method offers no solution, or rationale for a solution, for selecting among alternatives.
3. In this study’s analysis, only one of the two alternatives is included in Fig. 3. The other alternative would make it less intermediate in size. The selection of the more intermediate estimate is arbitrary at best, and the rationale for selecting one over others does not seem to be described in the methods.
In another example of ways the Template Method is problematic, the authors find the need to apply a “filter” to “limit the effect of variables with higher variance”. The logic behind this is unclear, and biases the outcome towards the outcome of moderate dimorphism.
In addition to major problems regarding the Template Method that must be addressed, this study has other major issues.
There are major questions about whether AL 333 assemblage can be treated as a “contemporaneous” (abstract) death assemblage. It is conclusively not a high-energy, catastrophic flooding event (Behrensmeyer, 2008). Additionally, MNI of the 333 assemblage may be as low as five individuals (Plavcan et al, 2005). This needs to be taken into consideration, especially given the sensitivity of the Template Method’s results to multiple representation of the same individual (Plavcan et al, 2005).
The analysis on page 7 does not test what the authors claim it tests. Computing the GMR using raw metrics and template ratios does not test whether TM is affect by scaling relationships between variables. The analysis in Suppl Table 4 uses the same variable, not two different variables. Once again, the issue of allometry makes the Template Method problematic and unreliable without knowing how allometric patterns might influence the analyses.
The attribution of KSD-VP-1/1 to A. afarensis is uncertain. The original article (Haile-Selassie et al, 2010, p 12121) notes that the “absence of cranial and dental elements imposes some restrictions on the specimen’s taxonomic assignment.” It is close to the transition between A. anamensis and A. afarensis. Most importantly, this is the only specimen from outside of Hadar, and Woranso-Mille has evidence of a species other than A. afarensis near this time period (Haile-Selassie et al, 2012).
The authors find that “Kadanuumuu template sample does predict greater dimorphism than does the Lucy template (Figure 2C).” Then the authors explain away this result and omit the results based on the Template Method applied to KSD-VP-1/1.
The results show that A. afarensis has higher dimorphism than chimpanzees. Table 2 shows that p< 0.05 whether using AL288-1 or KSD-VP-1/1 as templates. The results also show that A. afarensis is not significantly different from gorillas in most analyses. Why are these results not included in the conclusions?
In the end, the authors need to demonstrate how the Template Method can perform better than other methods, not with statistical tinkering and seemingly arbitrary data choices and omissions of analyses, but with modern taxa where relationships can be measured".

(WJ): This review recommended rejection and summarizes important issues raised in the other reviews, including

1. issues of allometry versus isometry
2, impact of estimates using different variables from the same template specimen
3. KSD-VP-1/1 has no associated craniodental remains, so it cannot be attributed with certainty to Au, afarensis (especially in view of multiple hominin taxa from Worsano-Mille).

The authors should not ignore or dismiss the important concerns raised in the review process. They should not ignore some of their own results. I am not sure that the authors can "fix" this study because the Template Method makes several demonstrably false assumptions and claims, but I believe they deserve a chance to address these criticisms head-on.

Reviewer 1 ·

Basic reporting

No comments.

Experimental design

No comments.

Validity of the findings

No comments.

Additional comments

The point of this paper is to compare the TM and GMM methods for estimating sexual dimorphism in Au. afarensis in an effort to explain previously published discrepancies yielded by these methods. Based on the standards outlined by PeerJ, I see no reason to reject this paper. The goal of the paper is clear and unambiguous and the authors demonstrate how the work fits with previous related research.

I am not particularly familiar with the technical details of the methodologies compared here and I can make no claim to understand them fully. Nevertheless, I do not feel that it is incumbent upon the authors to go into any more depth than they already do with respect to methodological explication. All of the methods utilized here are already established and have been discussed extensively in previous literature. Hence, I would simply note that I am not familiar enough with these methods to the point where I can definitively state that the conclusions of this paper are sound. I can only note that the conclusions are appropriately stated and connected to the original question noted in the Introduction. I would sincerely hope that this paper is also being reviewed by individuals who are more qualified to assess the technical content than I. Overall, the research presented here represents a new piece in an ongoing controversy that will likely continue to produce (on both sides) labyrinthine and somewhat tortuous papers that I would hope to avoid.

As a final minor editorial comment, I would ask the authors to please remember to place a comma after any prepositional phrase that begins a sentence. Example: "In most respects, the TM and GMM . . ."

Reviewer 2 ·

Basic reporting

Most parts of the manuscript are clearly written (but see methods section note below) and building the case against the GMR method is clear and concise. I had no real comments on the grammar or word choice. I will say that this debate has a long history, not simply between Reno et al. and Gordon et al., but also incorporating Plavcan et al. (2005) and Scott and Stroik (2006). I think it is reasonable to acknowledge this and briefly talk about the major findings and differences, and the cause of these differences.

Experimental design

The experimental design here is fairly simple and applies methods used in a number of previous analyses (Reno et al., 2003,2010, Gordon et al., 2008). It appears appropriate to their question. The description of the methods section could be improved (see below).
1.Line 153: The Sampling and Bootstrapping Procedures - this section could use some work at clarifying what is going on, particularly with regards to the reference samples. Based on previous works, I believe the reference samples are used in a similar fashion as the fossils - compared to the AL 288-1 and KSD templates, and SD is calculated from there, but this is very unclear.
2. I do wonder what would happen, rather than randomly selecting a new extant template for each iteration for the bootstrap comparisons of “biological error,” one template, such as a small, medium, and large extant individual was chosen (to mirror AL 288-1 and KSD) and the calculations were run using this one template for each randomly chosen sample of the comparative sample. Are the results the same given the changes in extant template size? In other words, is the pattern of dimorphism visible due to averaging over all possible templates rather than choosing one template as is done in the fossil analysis?

Validity of the findings

I have no reason to dispute the validity of Reno and Lovejoy’s findings - the procedure used is not complex. In addition to measuring dimorphism in Au. afarensis using the template method, the main thrust of their argument is that the method of GMR from Gordon et al. (2008) is biased by AL 288-1 and AL 128/129 contributing to the dimorphism ratios of numerous traits, which leads to larger ratios for more traits and therefore on average larger metrics of SD. Unless I am mistaken this is fairly clear. This is in direct opposition to Gordon et al. (2008) statement that though some individuals may contribute more than one trait (page 323) to the ratios calculated, this will not bias SD because they will contribute only one trait to each dimorphism ratio. The other part of their findings are that Au. afarensis has modern human like levels of dimorphism, but this is simply confirming their results using a bigger sample and a different template.

Additional comments

This paper is another in a series of works attempting to determine the level of sexual dimorphism in Australopithecus afarensis. It is well-written and lays out the argument simply. It builds on the work and approach of of Reno et al. (2003, 2005, 2010) applying what is known as the “Template Method” to an expanded Au afarensis fossil record. The issue this method and other similar approaches were designed to deal with is that few complete fossil individuals have more than a few traits in common. These methods attempt to answer the question of how to compare dimorphism across isolated skeletal elements. The recent discovery of KSD-VP-1/1 provides a new “template” in addition to AL 288-1 on which to compare isolated elements, and in this case the new fossil is a large probably male.
In the TM, ratios between isolated fossil measurements and those of one fairly complete template individual (AL 288-1 or KSD-VP-1/1 here). These size ratios are then standardized by multiplying them by the FHD of Lucy, or the Distal Tibial Breadth for KSD. CVs, and DBI were calculated based on these standardized ratios and compared to similarly computed ratios for extant great apes including modern humans.
The analysis also attempts to show that the recently developed Geometric Mean Method (Gordon et al., 2008) is biased by the number of skeletal elements contributed by single individuals, and that by lowering the # of elements AL 288-1 and AL 128/129 contribute to 1, dimorphism appears reduced to modern human levels using this method. The manuscript concludes that Au. afarensis had a similar level of dimorphism as modern humans.

Notes:
1. Line 56 “Singular comparisons of its largest individuals (333-3) with Lucy have led to a common impression that size dimorphism was pronounced in Au. afarensis. However, Lucy and AL 333 could differ in age by ten to hundreds of years, such that size variation may well reflect factors other than sexual dimorphism.” This is a fair point, though I would like to see a more recent citation for possible age difference.

2. But, more importantly, KSD-VP-1/1, which this paper points out is one of the largest Au. afarensis so far discovered (with a calculated femoral head diameter larger than AL 333-3 using the recent regression from Plavcan et al. 2014), is older than AL 288-1. If the above argument is that time change or other factors may have led to the observable substantial differences in size between AL 288-1 and AL 333-3 (for example), what is the reasoning here? I think the line cited above opens the analysis up to criticism - especially if the argument is made as it is in the paper that 333-3 and 288-1 were just ends of a continuum that makes up Au. afarensis

·

Basic reporting

The authors do a good job of providing their fossil data in supplemental tables, which I greatly appreciate. They also provide summary statistics for their comparative data. While I’d love it if they also provided individual measurements for their extant taxa, this is not the norm for our field and they are providing everything that is current practice in our discipline.

Experimental design

This manuscript is part of a continuing debate in the literature regarding sexual dimorphism in Australopithecus afarensis, and in part this manuscript is a response to some work published by myself and others (e.g., Gordon et al., 2008; Gordon, 2013). I’ve identified four areas below where I believe that there are either errors in the methodology used in this paper, or inaccurate representations of factors influencing differences between the results of the template method and the Gordon et al. analyses. These four areas are not all components of experimental design strictly-speaking, but I’ve grouped them here because they’re all related to each other.

1. Reliability of template method estimates

As has been pointed out elsewhere (e.g., Gordon et al., 2008; Gordon 2013), the template method assumes that all variables included in the analysis scale isometrically with each other. This is not a point where there should be any argument because it is a demonstrable mathematical fact. The template method first estimates values for a single reference variable (e.g., FHD or DSTB) using ratios between the values of that variable and all other available variables in a template specimen (e.g., Lucy or Kadanuumuu). Values for the reference variable are then estimated for other specimens by having their measurements for another variable that is also present in the template multiplied by the appropriate ratio. This assumes that the ratio between the reference variable and the other variable is the same in the template specimen and all other specimens. Another way of stating this is that shape (i.e., the ratio of the two variables) is constant across all specimens regardless of size, which is the definition of isometry. If the relationship between the reference variable and any other variable deviates from isometry or exhibits any variation about the isometric regression line, then this ratio technique will not produce accurate estimates. That observation is incontrovertibly true, and there isn’t much point in arguing over it. Where there is room for debate is over the questions of (1) whether the variables in a given data set scale perfectly isometrically with each other in A. afarensis, and (2) how much of an impact any deviations from isometry would have on a particular data set. The complete answers to these two questions are unknowable without measurements from several nearly complete skeletons (and the authors have not tested those questions, their assertions in the text notwithstanding), but one can address part of the second question by investigating the accuracy and precision of estimates of a reference variable in a single specimen using a series of other variables or different template specimens to generate the estimates.

One can get at this using the data provided by the authors in Supplemental Tables 1 and 2 (and I again want to express my appreciation that the authors included all of their fossil data in this submission). To begin with, there is one specimen which has an actual measurement for FHD for which FHD can also be estimated using Lucy as a template: A.L. 333-3. The actual measure of FHD for this specimen is 40.9, but it would be estimated as 38.2 using A.L. 333-3’s measure of FNKH and the ratio between FHD and FNKH in Lucy. That would underestimate the true FHD of this large individual by 6.6%. Of course, there is no way of knowing if other large individuals that don’t preserve FHD are similarly underestimated (and more broadly, if estimates are inaccurate in general).

If one then turns to specimens where actual FHD isn’t known (which prohibits us from investigating accuracy) but several estimates are possible using different ratios in Lucy as a template, one can evaluate precision. For example, A.L.322-1 is reported in Table 1 as having an estimated FHD of 32.2 based on CAPD. However, it would have an estimated FHD of 33.8 using HARB, and 27.9 using OLCB. That’s a difference of 5.9 between the maximum and minimum estimates, which is more than 19% of the midpoint of the range of these estimates. Shifting to DSTB estimates based on Kadanuumuu, DSTB in A.L. 128-1/129-1 is reported in Table 1 as estimated at 19.0 based on FLCL. However, it could also be estimated as 16.4 based on TMXT (which is smaller than the actual DSTB value of 17.2 in Lucy, and would act to increase estimated dimorphism in A. afarensis if used instead of the 19.0 estimate). Furthermore, if Lucy is used instead of Kadanuumuu as the template, five estimates of DSTB can be generated, ranging from 16.0 to 17.5, all of which are considerably smaller than the Kadanuumuu-based estimate. Similarly, A.L. 322-1 is reported in Table 1 as having an estimated DSTB of 19.7 using Kadanuumuu, but three estimates of DSTB ranging from 16.8 to 20.3 are produced using Lucy (a range that is 18.9% of the midpoint value).

Additionally, as the authors point out, the choice of reference variable and template is arbitrary. If one estimates FLCL in Lucy using either A.L. 128-1/129-1 or Kadanuumuu as a template, there are eight possible estimates ranging from 33.9 to 41.7, values whose range is 20.7% of the mean of the estimates (and 20.6% of the midpoint). This is a very large range of variation.

The point of all of the above is that the template method produces estimates that are imprecise at best, and that the data set that is used to calculate BDI or CV is highly dependent on which individual is selected as a template and which variables are used to estimate from; this would not be true if all variables scaled isometrically with each other across all specimens. Furthermore, there is no acknowledgment in this manuscript that different estimates are possible using different variables from the same template specimen for many of the specimens included in this analysis, nor any rationale for how the particular estimates that were used in this analysis were selected. This is a significant problem with the research design that needs to be addressed.

2. Sex assignment

In the discussion, the authors make much of the overlap in size between males and females in humans, chimpanzees, and bonobos, where they imply that mean method results are unreliable because (they argue) such methods assign sex to each individual, with presumed males above the mean and presumed females below the mean for any given variable. However, this is a red herring. The mean method does split a sample at the mean and calculate a ratio of the mean of the larger material to the mean of the smaller material, but it is recognized that this is not a direct measure of sexual dimorphism (which, if it were, I agree would imply that the members of the large group are all male and the members of the small group are all female), but rather one way to estimate relative size variation in the fossil sample (just as the coefficient of variation is another). It should be noted that BDI also splits the sample into specimens larger than a particular value and smaller than that value, then calculates the ratio of the means of the two groups. The only difference is that BDI iteratively calculates all such possible ratios where the number of individuals included in the group of larger specimens ranges from 1 to n-1 (and vice versa for the group of smaller specimens), where n is the total number of specimens, and then a weighted mean is calculated of these ratios. The point is that the ratios averaged by BDI also assume no overlap in size between the group of larger individuals and smaller individuals, just like the mean method. Not surprisingly, for a given set of univariate measurements both BDI and the mean method tend to give very similar results (e.g., see Fig. 1 in Gordon et al., 2008). Just like the coefficient of variation (CV), the mean method and BDI are measures of relative size variation that are correlated with sexual size dimorphism, but are not themselves measures of sexual size dimorphism. It is misleading to suggest that just because BDI and the mean method split a sample into two non-overlapping groups to calculate a measure of relative variation, then it follows that the use of either the mean method or BDI means that one assumes that there is no overlap in male and female size in the fossil sample. What is true is that the mean method and BDI tend to produces ratios that are very similar to actual dimorphism in extant taxa when males and females don’t overlap in size (e.g., in gorilla and orangutan postcranial measurements), and they produce ratios that are high relative to actual dimorphism when there is overlap between the sexes (e.g., humans, chimps, and bonobos) (Plavcan, 1994; Rehg and Leigh, 1999; Gordon et al., 2008). What is also true is that when extant samples with size overlap between sexes have mean method ratios that are high relative to actual sexual dimorphism ratios for those same samples, they are not elevated into the range of highly dimorphic taxa like gorillas and orangutans (Gordon et al., 2008; Gordon, 2013), so it is not accurate to suggest that gorilla-like mean method ratios for A. afarensis are simply the product of an incorrect assumption that there is no overlap in size between the sexes in A. afarensis.

3. The number of times specimens are represented in various methods

The authors argue in multiple places that the geometric mean method of Gordon et al. (2008) places undue weight on specimens that are more complete than other specimens through “multiple inclusion” of such specimens. This assertion of undue influence is incorrect, and shows a fundamental misunderstanding of a key difference between the template method and Gordon et al.’s method.

Both methods incorporate information from multiple variables into an estimate of relative size variation using two steps of estimation, although the steps differ for the two methods. The template method first estimates values for a single reference variable using presumed isometric scaling relationships among all variables, then estimates relative variation in the resulting estimated data set using whatever preferred method one cares to use (e.g., BDI, mean method, or CV). Gordon et al.’s method first estimates relative variation in all variables separately, including only actual measurements (as opposed to estimated), using one of the relative size variation methods (again, it could be BDI, mean method, or CV), and then estimates overall relative size variation in the data set as the geometric mean of those estimates of relative variation in each of the individual variables. So if the Gordon et al. analysis had a similar first step to the template method, then in using all measurements from Lucy it absolutely would create a cluster of highly similar estimates for a single reference variable that would unduly affect the subsequent measure of relative variation. However, the Gordon et al. analysis doesn’t have such a step. Instead, each specimen is included exactly once in the calculation of relative size variation for every variable in which it appears. It would be highly misleading to calculate measures of size variation for both FHD and HHD in which Lucy was included in one but not the other; Lucy really does have both a femoral head and a humeral head. To calculate a mean of ratios for FHD and HHD in which Lucy is represented in both ratios is not counting that specimen twice, but rather combining estimates of size variation for different variables in which every specimen that has a measurement for that variable is included exactly once.

4. Changing Lucy

The authors also include analyses where Lucy is increased in size and the influence of Lucy and A.L. 128-1/129-1 on overall size variation is evaluated. I agree that if Lucy were made 30% larger, or if Lucy and A.L. 128-1/129-1 were removed, A. afarensis would appear much less variable. However, Lucy and A.L. 128-1/129-1 are the size they are. The authors seem to be arguing that because these are the two smallest specimens in the sample, their effect on size variation should be discounted. That same argument could be used to remove from the analysis all specimens that deviate from the mean. The fact is that these two specimens do exist and contribute to the overall variation present in the sample, and that can’t be argued away.

Citations:
Gordon AD. 2013. Sexual size dimorphism in Australopithecus: current understanding and new directions. In: Reed, K. E., Fleagle, J. G. & Leakey, R. E. (eds.) The Paleobiology of Australopithecus. Dordrecht: Springer.
Gordon AD, Green DJ, Richmond BG. 2008. Strong postcranial size dimorphism in Australopithecus afarensis: results from two new resampling methods for multivariate data sets with missing data. Am J Phys Anthropol, 135, 311-28.
Plavcan JM. 1994. Comparison of 4 simple methods for estimating sexual dimorphism in fossils. Am J Phys Anthropol 94:465-476.
Rehg JA, Leigh SR. 1999. Estimating sexual dimorphism and size differences in the fossil record: a test of methods. Am J Phys Anthropol 110:95-104.

Validity of the findings

Even if one were to accept that the methods used here were sound (see my comments above), there is still a significant problem with the interpretation of results. Just as in previously published versions of this work, the authors choose to focus on the result that their measure of postcranial relative size variation does not significantly differ between A. afarensis and modern humans, arguing that therefore A. afarensis had modern human-like dimorphism (and by inference, a modern human-like level of sexual selection). However, if one examines the template method results, Table 2 shows that for BDI and CV, A. afarensis is either not significantly different from, or has significantly greater values than, chimpanzees, and in six out of eight tests is not significantly different from gorillas (the values in Table 2 can be converted to one-tailed p-values by dividing the number of resampled extant values that at least as extreme as the A. afarensis value by 1000, the total number of resampled values). The authors argue that the small samples (i.e., all samples except the Combined Afar sample using the Lucy Template) produce relatively wider distributions for comparative extant taxa than larger samples (which is true), and as such those results (which do not allow them to differentiate A. afarensis values from the means for humans, chimpanzees, or gorillas) should be downplayed in favor of the results based on the largest sample.

I disagree that those other results should be set aside as uninformative due to small sample size, because the authors are trying to make arguments about the influence (or lack thereof) of using Kadanuumuu as the template specimen and the differences between analyses based on different templates are almost certainly not due solely to sample size differences. But for the sake of argument I’ll focus only on the results they highlight, those of the Lucy-based Combined Afar sample. While the authors are not explicitly making the incorrect argument that failure to reject the null hypothesis means that the null hypothesis is true when they state the level of dimorphism in A. afarensis is human-like, they are implying that flawed argument. But even if that observation is correct, they have not adequately explained how to interpret a level of variation that is simultaneously not significantly different from humans and greater than that seen in chimps. Decades of work in primates has shown that high levels of mass dimorphism are associated with social structures that allow for high degrees of sexual selection, and chimpanzees are both more mass dimorphic and subject to more intense sexual selection than modern humans. The problem is that humans have unusually high skeletal dimorphism compared to their mass dimorphism, so that they are more skeletally dimorphic, but less mass dimorphic than chimpanzees. The authors provide a potential explanation for why this might be based on sex differences in tissue distributions in humans, but this doesn’t address the problem that a human-like level of skeletal dimorphism that is also significantly greater than chimpanzee skeletal dimorphism is equally strong support for a level of sexual selection that is higher than that seen in chimpanzees (and thus also higher than in humans) as it is for a lower level of sexual selection similar to that seen in modern humans. At the very least, the authors should acknowledge this point in their work.

Additional comments

Leaving aside for the moment my criticisms above and addressing the rest of the manuscript, although I don’t agree with much of the inference and interpretation presented in the discussion and elsewhere (and I don’t want to extend an already overly-long review to address these things point by point), much of this boils down to differences in interpretation, which in my book is not a reason to argue against the publication of something that is otherwise technically sound. I’m not convinced by all of the arguments presented on the last couple of pages of this manuscript, but they introduce some interesting ideas that I would have liked to see fleshed out a little bit more and that would be interesting to have a conversation about.

Reviewer 4 ·

Basic reporting

The basic reporting is fine. The manuscript is well written and clearly lays out the problem, the approach, the results, and the interpretations.

Experimental design

The submission meets PeerJ's requirements in the following ways: it describes original primary research, the research question is clearly defined, the methods are sufficiently described, and there are no apparent ethical issues.

The submission falls short in that its statistical approach and the interpretations of the results are flawed. I outline the problems in much greater detail in my comments to the authors below.

Validity of the findings

As outlined below in my comments to the authors, I am not convinced that their results are robust and statistically sound. There are some major issues that they need to address before the submission can be published.

Additional comments

Review of “From Lucy to Kadanuumuu: Balanced analyses of Australopithecus afarensis assemblages confirm only moderate skeletal dimorphism in the species” by Reno and Lovejoy

Summary of the manuscript

This manuscript is the latest in a series of papers by the authors and their colleagues dealing with the problem of sexual dimorphism in Australopithecus afarensis. Prior to their first publication on the subject in 2003, several studies had reached the conclusion that size variation within the skeletal and dental samples assigned to this early hominin was high, suggesting a level of body size dimorphism at least as great as that seen in the most dimorphic living hominoids (gorillas and orangutans). The analyses conducted by Reno, Lovejoy, and colleagues, including the one under consideration here, challenge this view, with these authors interpreting their results as indicating a much lower, humanlike pattern of dimorphism.

The current contribution is novel in two respects. First, it includes material from the recently discovered KSD-VP-1/1 partial skeleton, referred to as “Kadanuumuu,” which appears to represent a large and presumably male individual of A. afarensis and expands the range of size variation in an important subset of the sample for this species. Second, and more important, the authors attempt to reconcile their results, which have remained consistent since 2003, with those of a study that was critical of their work, conducted by Gordon, Green, and Richmond (2008), who reached the conclusion that A. afarensis was much more dimorphic than extant humans.

Reno and Lovejoy conclude after a series of analyses that their methodological approach and the one taken by Gordon et al. should produce similar results. They then evaluate a number of explanations for the conflict in the case of A. afarensis. The one they favor is that Gordon et al.’s analysis includes multiple measurements from two of the smallest individuals of A. afarensis discovered to date—A.L. 288-1 (the famous “Lucy”) and A.L. 128/129—and that this multiple representation at the small end of the size range biases the results of the analysis toward finding high dimorphism, because it gives these very small individuals undue influence on the results.

Major criticisms

Multiple representation of single individuals and sample composition have been persistent problems in the debate over dimorphism in A. afarensis since Reno, Lovejoy, and colleagues published their first paper on the subject in 2003. One of the criticisms directed at Reno, Lovejoy, and colleagues’ work pertains to the way in which they treat a subset of the A. afarensis sample from the A.L. 333 site. Most paleoanthropologists, including Reno and Lovejoy, recognize that the 26 fossil specimens from A.L. 333 were contributed by far fewer individuals. The exact number is not known. Reno and Lovejoy assume n = 9 for their analyses, which is the minimum number of individuals (MNI) estimated from dental remains at the site. However, as pointed out by Plavcan and colleagues in their 2005 response to Reno, Lovejoy, and colleagues’ 2003 paper:

“The strict MNI for the postcranial elements measured by Reno et al. is only three (the maximum number of duplicated parts) for adults, and four including the subadult specimens used by
Reno et al. The actual number of individuals represented by postcrania is, of course, unknown, but is almost certainly greater than four. However, reasonably clear associations between many of the A.L. 333 postcranial elements (some of the Reno et al. specimens actually appear to be antimeres) suggest to us that the postcranial sample used by Reno et al. comprises between five and eight individuals” (Plavcan et al., 2005, p. 314; this reference is conspicuously absent from Reno and Lovejoy’s reference list, though they do mention that “previous simulations sampling from as few as 5 individuals to represent A.L. 333 did not substantively alter our results”; see lines 171–172).

So, on the one hand, Reno and Lovejoy criticize Gordon et al. for including multiple specimens from the same two individuals in their analysis, but on the other hand Reno and Lovejoy include multiple specimens from the same individuals in their analysis. Why is there a double standard here? If Gordon et al.’s treatment of A.L. 288-1 and A.L. 128/129 is confounding their analysis, why doesn’t the inclusion of many elements from a small number of individuals at A.L. 333 confound Reno and Lovejoy’s analysis? This difference in treatment gives the impression that the authors are trying to have their cake and eat it too. Also, if there isn’t a substantive difference between modeling the A.L. 333 specimens as coming from 5 individuals vs. 9, then why not take the more conservative approach and model it as containing 5? By using 9, the authors are opening themselves up to critics who have used this issue to challenge Reno, Lovejoy, and colleagues’ results previously.

Related to the issue raised above, Reno and Lovejoy conduct an interesting exercise to demonstrate the effect of A.L. 288-1 and A.L. 128/129. They limit both of these specimens to contributing a single measurement each to the estimate of dimorphism in A. afarensis (i.e., 2 measurements instead of 10) and find that this lowers the estimate of dimorphism. This is similar in some respects to an exercise conducted by Scott and Stroik (2006) in their analysis of the effect of reducing the data contributions of the A.L. 333 individuals. Scott and Stroik found that limiting the A.L. 333 contributions resulted in an estimate of size variation that could not be statistically distinguished from chimpanzees, humans, or gorillas. Why haven’t the authors of the present manuscript conducted a similar sort of exercise for A.L. 333? If Gordon et al.’s sample deserves that level of scrutiny, then why not Reno and Lovejoy’s? Not doing so leaves the reader with the impression that the authors are not approaching the two data sets in a balanced way. As noted by Reno and Lovejoy, Gordon et al. (2008) conducted analyses in which they constrained the smallest two individuals in their bootstrap samples to contribute specimens in a manner identical to A.L. 288-1 and A.L. 128/129. The results of those analyses differed in some ways from the other (unconstrained) analyses, but the overall signal was not qualitatively different in that it indicated substantial dimorphism in A. afarensis. In modeling A.L. 288-1 and A.L. 128/129 in this way, Gordon et al.’s approach is similar in some respects to Reno and Lovejoy’s approach to modeling the A.L. 333 sample. So why is the latter approach okay but not the former?

Reno and Lovejoy conduct a separate analysis that excludes the A.L. 333 specimens, which allows them to avoid some of the difficult issues discussed above. The authors describe these results as follows: “The smaller non-333 sample also shows a similar pattern (Figure 3B), although it just fails to reach statistical significance at p = 0.05 level for both chimpanzee (low dimorphism) and gorillas (high dimorphism) (Table 2).” The p-values in question range from 0.073 to 0.084, and they are all one-tailed. Why? What a priori reason do the authors have for conducting one-tailed tests rather than two-tailed tests? In the case of the comparison with gorillas, one reason might be that the authors think the gorilla represents some kind of upper limit on dimorphism in primates. If that is the authors’ view, then they should make that explicit. However, they’d still need to address the work of Jay Kelley and colleagues suggesting that greater levels of dimorphism are possible and in fact characterize living (e.g., papionins) and extinct (e.g., Lufengpithecus and other Miocene apes) primates. Given that possibility, it seems that a two-tailed test is most appropriate in these sorts of comparisons. Without a robust justification for using a one-tailed test, the authors are violating the rules of null hypothesis significance testing (which apply to resampling methods), once again opening themselves up to criticism. The authors are not alone in misusing one-tailed tests in this context; this practice is common among biological anthropologists, but that doesn’t mean it’s correct. Doubling the p-values in Table 2 to approximate two-tailed values renders the statistical results far less supportive of the authors’ preferred interpretations.

There are also problems related to the interpretation of the statistical results. Strictly speaking, the authors are conducting tests of equal relative variation between samples. A significant difference means that the relative variation is not equal, whereas a nonsignificant result means that a difference in relative variation cannot be detected. The authors interpret a significant result as indicating a difference in sexual dimorphism, whereas a nonsignificant result indicates a similar level of dimorphism. Such interpretations are reasonable, but they are not the only interpretations. For example, one criticism of Reno, Lovejoy, and colleagues’ work that has been raised by multiple critics is that the low level of size variation in Reno, Lovejoy, and colleagues’ sample might reflect an unbalanced sex ratio. Previous studies (e.g., Plavcan, 1994, AJPA) have shown that, where the sex of specimens is uncertain (as is the case here), measures of size variation such as the CV can underestimate the true level of sexual dimorphism. So, when the value for a fossil sample of unsexed individuals is compared to a distribution generated by bootstrapping from a sex-balanced sample of gorillas, and the fossil value is found to fall at the lower end of the resulting gorilla distribution, does that mean the fossil species had a lower level of dimorphism, or does it mean that the fossil sample is sex-biased?

It has been argued that the bootstrap accounts for sex bias because not all of the individual bootstrap samples (or even most of them) will be sex-balanced. But as any introductory text on resampling methods will tell you, bootstrap distributions reflect the statistical properties of the sample that was used to generate the distribution. If the original sample is sex-balanced, then the bootstrap distribution will reflect that balance. Therefore, such bootstrap distributions cannot account for sex bias. This is an important point, because in the type of analysis conducted by Reno and Lovejoy, the fossil value is compared to the distribution; the sex composition of any single bootstrapped sample matters little, because it is the entire distribution and its statistical properties that are used in the statistical evaluation of the fossil value. In order to directly examine the effect of sex bias, the bias needs to be built into the resampling protocol. (Note that this also applies to modelling the contributions of the individuals in the A.L. 333 sample; the effect in the number of individuals in the sample needs to be examined explicitly, even if not all of the 9 bootstrapped individuals contribute to the calculation of dimorphism in each iteration of the procedure.)

The potential for sex-biased fossil samples to mislead our attempts to estimate sexual dimorphism in fossil hominins is illustrated very well in Lockwood and colleagues’ 2007 analysis of Paranthropus robustus in Science. If those authors are correct, then the P. robustus sample is heavily biased towards male individuals, a situation that has led to underestimates of dimorphism in that species. Lockwood and colleagues develop a scenario that might explain why the P. robustus sample is biased, which is worth consulting, but sex-biased samples can result from random processes as well. Whether or not the A. afarensis sample, or parts of it, is sex-biased is obviously not known, but the possibility that it is, and that the bias affects estimates of size variation, and therefore estimates of sexual dimorphism, needs to be confronted in a substantive way.

Other comments

Reno and Lovejoy treat the measures of variation for the A. afarensis sample as point estimates, free of error and uncertainty. This practice is typical for this sort of analysis, usually because fossil samples are so small that confidence intervals are meaningless. However, in this case, Reno and Lovejoy’s sample is not small; in fact, it is quite large, and the authors discuss the importance of maximizing fossil samples. The descriptor “large” applies even to the subsets of the sample; with an n of 16, the non-A.L. 333 sample size is enormous in terms of what paleoanthropologists normally deal with. So why not also bootstrap the A. afarensis sample and its various subsets to create distributions that can be compared to the distributions for humans, chimps, and gorillas, like Gordon et al. did in their analysis? That approach would be more conservative and therefore more statistically robust.

The authors state in lines 298–302: “However, mass and skeletal dimorphism do not have the same relationships in humans and apes, making the task of inferring mass dimorphism from skeletal dimorphism difficult for early hominids (Reno et al., 2003; Gordon et al., 2008; Plavcan, 2012b). Instead, we have avoided this conundrum by simply focusing on skeletal dimorphism, an important indicator of total size and sex differences in growth (Hamada and Udono, 2002; Reno et al., 2003).”

And lines 355–356: “the three most closely related living taxa to Au. afarensis, humans, chimpanzees, and bonobos, all show substantial overlap between the sexes.”

Furthermore, in lines 361–362: “Appreciable overlap between the sexes would suggest an absence of targeted selection for increased male size.”

Chimps and bonobos have lower skeletal dimorphism than humans but higher body mass dimorphism. Chimps, bonobos, and humans all differ in their reproductive strategies. Given that we estimate body mass from the skeleton in fossil species, doesn’t this mean that we can’t really infer anything about A. afarensis reproductive strategies with any degree of confidence?

---

## Round 0.2 · accepted · Accept

· Academic Editor

Accept

Thank you for your thoughtful and detailed response to reviewer concerns and suggestions. I think a stronger, better argued paper has emerged along the way. Thanks too for keeping the tone respectful and professional in your rebuttal. I'm not sure this is the last word on this debate, but it certainly advances the discussion in a most positive way.